# New Horizons in Metastatic Colorectal Cancer: Prognostic Role of CD44 Expression

**DOI:** 10.3390/cancers15041212

**Published:** 2023-02-14

**Authors:** Pina Ziranu, Valentina Aimola, Andrea Pretta, Marco Dubois, Raffaele Murru, Nicole Liscia, Flaviana Cau, Mara Persano, Giulia Deias, Enrico Palmas, Francesco Loi, Marco Migliari, Valeria Pusceddu, Marco Puzzoni, Eleonora Lai, Stefano Cascinu, Gavino Faa, Mario Scartozzi

**Affiliations:** 1Medical Oncology Unit, University Hospital and University of Cagliari, SS 554 km 4500 Bivio per Sestu, Monserrato, 09042 Cagliari, Italy; 2Division of Pathology, Department of Medical Sciences and Public Health, AOU Cagliari, University of Cagliari, 09124 Cagliari, Italy; 3Department of Medical Oncology, Università Vita-Salute, San Raffaele Hospital IRCCS, 20019 Milan, Italy

**Keywords:** metastatic colorectal cancer, biomarkers, CD44 expression, cancer stem cells

## Abstract

**Simple Summary:**

In our study, we analyzed the role of CD44 expression as a prognostic marker in patients with metastatic colorectal cancer. CD44 is a multi-structural and multifunctional non-kinase transmembrane glycoprotein. Its expression is recognized as a molecular marker of cancer stem cells. Due to its role in the maintenance of stemness and the function of CSCs in tumor progression, CD44 could be an important prognostic marker. In our study, elevated CD44 expression correlated with more aggressive tumor behavior and poor prognostic features, suggesting its role as a novel prognostic marker and potential therapeutic target for mCRC patients.

**Abstract:**

Background: The transmembrane glycoprotein CD44, the major hyaluronan (HA) receptor, has been proven to regulate cell growth, survival, differentiation, and migration. It is therefore widely considered to be involved in carcinogenesis. Its role as a new therapeutic target in solid tumors is under evaluation in clinical trials. The prognostic value remains controversial. Here, we aimed to investigate the correlation between CD44 expression and the clinicopathological features and survival in metastatic colorectal cancer (mCRC) patients. Methods: Data from 65 mCRC patients of the Medical Oncology Unit, University Hospital and University of Cagliari were retrospectively collected from 2008 to 2021. Immunohistochemical analysis was performed at the Pathology Division, University Hospital of Cagliari on 3 μm thick sections obtained from paraffin blocks. The intensity of immunohistochemical staining was subclassified into four groups: score 0 if negative or weak membrane staining in less than 10% of tumor cells; score 1+ if weak membrane staining in at least 10% of tumor cells or moderate membrane staining in less than 10% of tumor cells; score 2+ if moderate membrane staining in at least 10% of tumor cells or intensive membrane staining in less than 10% of tumor cells; score 3+ if intense membrane staining in at least 10% of tumor cells. Based on this score, we distinguished patients into low CD44 expression (score 0, 1+, 2+) and high CD44 expression (score 3+). Statistical analysis was performed with MedCalc (survival distribution: Kaplan–Meier; survival comparison: log-rank test; association between categorical variables: Fisher’s exact test). Results: Patients’ median age was 66 years (range 49–85). Regarding CD44 expression, score was 0 in 18 patients, 1+ in 15 patients, 2+ in 18 patients, and 3+ in 14 patients. Median overall survival (mOS) was 28.1 months (95%CI: 21.3–101). CD44 overexpression (3+) was correlated with poor prognosis (*p* = 0.0011; HR = 0.2), with a mOS of 14.5 months (95%CI 11.7 to 35.9) versus 30.7 months (95%CI 27.8 to 101) in lower CD44 expression. Higher CD44 expression was associated with clinically poor prognostic features: age ≥ 70 years (*p* = 0.0166); inoperable disease (*p* = 0.0008); stage IV at diagnosis (*p* = 0.0241); BRAF mutated (*p* = 0.0111), high-grade tumor (*p* = 0.0084). Conclusions: CD44 markedly correlated with aggressive tumor behavior and contributed to the earlier progression of disease, thus suggesting its role as a novel prognostic marker and potential therapeutic target for mCRC patients.

## 1. Introduction

GLOBOCAN epidemiological data show nearly 2 million new colorectal cancer (CRC) cases worldwide in 2020, accounting for about 10% of all cancer cases. It represents the third most common neoplasm in men and the second most common in women [1]. Despite advances in surgical techniques, chemotherapy, and radiotherapy, CRC remains the second leading cause of cancer death, accounting for nearly 1 million deaths annually [1]. Among new CRC diagnoses, 20% of patients have metastatic disease at presentation, and up to 25% who have been diagnosed as having localized disease will later develop metastases. The 5-year survival for stage IV disease is only 12% [2].

Over the last decade, the management of CRC patients changed considerably, particularly in metastatic disease, mainly due to the introduction of combination chemotherapy with targeted agents, leading to more curative resections and prolonging survival in patients with unresectable disease. A deeper understanding of the tumorigenesis mechanisms facilitated tumor characterization, prognosis, and patient stratification, with a view to personalized medicine. Indeed, like other solid tumors, CRC is a heterogeneous disease in which different subtypes can be distinguished by their specific clinical and/or molecular features [3,4,5,6]. Therefore, understanding the molecular pathways underlying the initiation and development of CRC is essential to identify new biomarkers for diagnosis and prognosis, thereby improving outcomes.

Ongoing efforts to investigate new treatment strategies for CRC include understanding the involvement of cancer stem cells (CSCs) [7]. Greater knowledge of the CSCs’ metabolic characteristics is undoubtedly related to discovering a subpopulation of tumor-initiating cells in CRC. Indeed, in 2007 the existence of tumorigenic and non-tumorigenic cells within CRC was demonstrated, implying that not all cells within a tumor are capable of initiating and sustaining the neoplastic growth [8]. This concept has important therapeutic implications and suggests that targeting CSCs is necessary to increase the efficacy of therapeutic strategies. CSC features include the ability to self-renew, resistance to chemotherapy and/or radiotherapy, and increased metastatic potential that the tumor microenvironment can shape [9].

Specific surface markers of CSCs appear to have a prognostic role [10,11,12]. However, their clinical relevance in CRC is currently controversial and has been reported only in small institutional studies [12,13,14,15,16,17,18,19,20,21,22]. Among stem cell surface markers, CD44 is of great interest. Indeed, in vitro studies have shown that a single CRC cell expressing CD44 can produce highly heterogeneous CRCs [23,24].

CD44 is a multistructural and multifunctional non-kinase transmembrane glycoprotein expressed in embryonic stem cells and at various levels in other cell types, including connective tissues and bone marrow [25,26]. The expression of CD44 is recognized as a molecular marker of CSCs [27]. It is encoded by a single gene containing 19 exons. The first five and last five exons are constant and encode the shortest isoform of CD44 (85–95 kDa) called standard CD44 (CD44s). Variant isoforms (CD44v) are generated by alternative splicing and possess the ten constant exons and any combination of the remaining nine variant exons [28,29]. The main ligand of CD44 is hyaluronic acid (HA), an abundant component of the extracellular matrix (ECM) expressed by stromal and tumor cells [30]. The HA-CD44 complex induces conformational changes by binding adaptor proteins or cytoskeletal elements to the intracellular domain. This binding activates several signaling pathways that lead to cell proliferation, adhesion, migration, and invasion [31,32].

Therefore, the role of CD44 in maintaining stemness and the CSC function in tumor progression suggests that CD44 might be an important prognostic marker. Here, we aimed to investigate the correlation between CD44 expression and the clinicopathological features and survival of metastatic colorectal cancer (mCRC) patients.

## 2. Materials and Methods

We retrospectively collected data from 65 cases of mCRC patients diagnosed between 2008 and 2021, referred to the Medical Oncology Unit of the University Hospital of Cagliari. Baseline demographic and clinical characteristics, treatment, and survival information were collected from clinical charts. Pathological and molecular features were retrieved from histological reports. The following data were collected: gender, age, Eastern Cooperative Oncology Group (ECOG) performance status (PS) at diagnosis of metastatic disease, the onset of metastatic disease, primary tumor location, sites of metastases, mucinous histology, grade of differentiation, CD44 and CDX2 tumor expression, BRAF/RAS mutational status, MSI/MMR status, and treatment outcome. For study purposes, right-sided and left-sided CRC primary tumors were defined as proximal or distal to the splenic flexure. Ethics Committee approval was obtained for the study (Protocol number 2020/10912—code: EMIBIOCCOR) and written informed consent was obtained from all participants for their tissues to be utilized for this work.

Tumor samples were retrospectively tested for CD44 immunohistochemical (IHC) expression with the aim of evaluating the correlation with clinical outcome in terms of overall survival (OS), progression-free survival (PFS), response rate (RR), disease control rate (DCR). The primary endpoint was the OS rate at 24 months. The secondary endpoints were overall survival (OS), progression-free survival (PFS), objective response rate (ORR), and disease control rate (DCR) in the first and second line.

### 2.1. Histological, Immunohistochemical, and Molecular Analysis

Tumor samples were routinely processed for histological observation and stained with haematoxylin–eosin (H&E). For immunohistochemical analysis, 3 μm thick sections were obtained from the paraffin block. All reagents were purchased from Ventana Medical Systems Inc. 1910 E. Innovation Park Drive Tucson, Arizona 85755 USA. The sections were automatically dewaxed and rehydrated with EZ Prep 1X (Ref. 950–102) and pre-treated with heat-induced epitope retrieval in Ultra CC1 (Ref. 950–224), following the manufacturer’s instructions. Slides were then incubated at room temperature with anti-human CD44 rabbit monoclonal antibody—clone SP37 (Ref. 790–4537) and with anti-human CDX2 rabbit monoclonal antibody—clone EPR2764Y (Ref. 760–4380). All immunostaining procedures were performed using the UltraView Universal DAB Detection Kit (Ref. 760–5000) on the BenchMark Ultra (Ventana Medical Systems Inc. 1910 E. Innovation Park Drive Tucson, Arizona 85755 USA) instrument, according to the manufacturer’s instructions. For CD44 interpretation, we used the following grading score system, based on the HER2/neu scheme (Table 1) (22). Based on this score, we distinguished patients as low CD44 expression (score 0, 1+, 2+) and high CD44 expression (score 3+). The presence of intense membrane staining in at least 10% of tumor cells is classified as score 3+ (Figure 1). RAS and BRAF gene mutational status was assessed by pyrosequencing of formalin-fixed, paraffin-embedded (FFPE) archival tumor tissue samples from primary tumors or metastases. Expression of MMR proteins (MLH1, MSH2, MSH6, and PMS2) was measured by immunohistochemistry. Histological, immunohistochemical, and molecular analysis were conducted at the Division of Pathology of the University Hospital of Cagliari.

### 2.2. Statistical Analysis

Statistical analysis was performed with the MedCalc® Statistical Software version 20.008 (MedCalc Software Ltd, Ostend, Belgium; https://www.medcalc.org; 2021; accessed on 1 June 2022). The association between categorical variables was estimated by the Fisher exact test for categorical binomial variables or by the chi-square test in all other instances. Survival probability over time was estimated by the Kaplan–Meier method. Significant differences in the probability of survival between the strata were evaluated by the log-rank test. The independent role of variables that were statistically significant in a univariate analysis was assessed with a logistic regression analysis. OS was defined as the time interval between the date of the beginning of the first-line treatment to death or the last follow-up visit for patients who were lost to follow-up. PFS-I was defined as the interval between the date of the beginning of the first line of treatment to death, first sign of clinical progression, or the last follow-up visit for patients who were lost to follow-up. PFS-II was defined as the interval between the date of the beginning of the second line of treatment to death, first sign of clinical progression, or the last follow-up visit for patients who were lost to follow-up. ORR was defined as the percentage of patients who achieved a partial or complete response to treatment according to RECIST version 1.1. DCR was defined as the percentage of patients with stable disease or partial/complete response to treatment.

In order to detect a difference in the effect size with statistical significance in the proportion of patients alive at 24 months according to CD44 status and assuming a 24-month OS of 65% in the low CD44 expression group and 25% in patients with high CD44 expression, at least 52 patients were necessary with α = 0.2 and β = 0.2, using a “comparison of proportion test.” A *p*-value <0.05 was considered statistically significant.

## 3. Results

Data from 65 mCRC patients diagnosed between 2008 and 2021, referred to the Medical Oncology Unit of the University Hospital of Cagliari, were collected. Patients’ characteristics were consistent with a stage IV CRC population (Table 2). The median age was 66 years (range, 49–85), 37 were male (57%) and 28 were female (43%). Negative or weak CD44 membrane staining in less than 10% of tumor cells (score 0) was observed in 18 (27.7%) patients, 15 (23.1%) showed weak membrane staining in at least 10% of tumor cells or moderate staining in less than 10% of tumor cells (score 1+), 18 (27.7%) showed moderate membrane staining in at least 10% of tumor cells or intense staining in less than 10% of tumor cells (score 2+), and 14 (21.5%) showed intense membrane staining in at least 10% of tumor cells (score 3+).

As of the data cut off, 31 December 2021, 13 (20%) patients were alive, while the remaining 52 (80%) had died.

### Clinical Outcomes

At a median follow-up of 26.6 months (95% confidence interval (CI) 21.4 to 28.8), the median OS was 28.3 months (95%CI 21.3 to 101). The proportion of patients alive at 24 months was 70% in the low CD44 expression group versus 21% of high CD44 expression patients (*p* = 0.0001) (Figure 1). Median OS was 30.7 months (95%CI 27.8 to 101) in patients with low CD44 expression versus 14.5 months (95%CI 11.7 to 35.9) in the high CD44 expression group (*p* = 0.0001) (Figure 2).

Patients involved in our analysis received first- and second-line chemotherapy treatment based on the patient’s clinical features, tumor-related biological characteristics, and clinician preference (Table 2). There is substantial uniformity between the populations with low and high CD44 expression. However, it is useful to note a trend towards greater first-line triplet use in highly expressed CD44. In low CD44 expression, the chemotherapy regimen used (monotherapy, doublet, or triplet) did not impact first-line (*p* = 0.3) or second-line treatment (*p* = 0.4). The same applies to highly expressed CD44 (*p* = 0.3 and *p* = 0.1, respectively). The biological drug used (anti-VEGF, anti-EGFR, or nothing) in low CD44 expression also did not impact the first line (*p* = 0.2) nor the second line (*p* = 0.3). The same applies to highly expressed CD44 (*p* = 0.6 and *p* = 0.3, respectively).

Furthermore, regarding first- and second-line responses in terms of PFS and RR, our study showed no statistically significant difference in patients with high and low CD44 expression. However, we can see a trend towards higher treatment efficacy in patients with low CD44 expression compared to those with high expression (Table 3).

We analyzed the impact of different clinicopathological features on OS and TTP. In the univariate analysis, it was interesting to note that clinicopathological features associated with poor prognosis were more frequent in patients with highly expressed CD44 (Table 4). We recorded an age ≥70 years in 42.8% of patients with highly expressed CD44 versus 23.5% of those with low CD44 expression (*p* = 0.0166). Of patients with highly expressed CD44, 58.3% had an inoperable disease at diagnosis versus 9.8% of those with low CD44 expression (*p* = 0.0008). Furthermore, at diagnosis, 92.9% of patients with high expression of CD44 presented with stage IV versus 60.8% of patients with low CD44 expression (*p* = 0.0241). Of patients with highly expressed CD44, 57.2% had a grade 3 tumor compared with 19.6% of patients with low CD44 expression (*p* = 0.0084). Finally, BRAF was mutated in 9.8% of patients with non-overexpressed CD44 compared to 35.7% of patients with CD44 3+ (*p* = 0.0111).

## 4. Discussion

Our study analyzed the correlation between CD44 expression and clinically relevant CRC outcomes using a large cohort of patients. CD44 is a cell surface glycoprotein involved in cell–cell interaction, adhesion, and migration [33]. It participates in multiple physiological processes, and its aberrant expression and dysregulation contribute to tumor initiation and progression [34]. The different functional roles of CD44s and specific CD44v still need to be fully elucidated. CD44 and the various CD44v isoforms have partly overlapping and distinct functional roles. CD44v isoforms have additional binding sites that promote the interaction of CD44 with molecules in the microenvironment. CD44v can act as co-receptors by binding/sequestering growth factors on the cell surface and presenting them to their specific receptors [35,36]. Moreover, CD44v has been reported to be expressed in metastatic tumors [37,38], whereas switching between CD44v and CD44s may regulate epithelial to mesenchymal transition (EMT). CD44 can undergo isoform switching in cancer cells [39]. Recently, it has been seen that in pancreatic cancer cells, EMT phenotype induction seems to require the upregulation of CD44 expression with isoform switching of CD44v to CD44s expression [40].

Cells with CD44 overexpression appear to have less spontaneous apoptosis and are more resistant to drug-induced cell death, inducing an anti-apoptotic effect. High expression of CD44 confers a selective advantage to resist apoptosis, thereby promoting cell transformation into a malignant phenotype in conjunction with other anti-apoptotic factors [41,42,43,44].

Recent studies have shown that CD44 is the most common CSC surface marker and plays an essential role in communication between CSCs and the tumor microenvironment [45]. CSCs represent a cohort of long-lived cells that can generate lifelong cell progeny and fuel tumor growth [46,47]. Their existence may explain the ineffective response of many therapies, leading to poor prognosis and recurrence in patients with different types of cancer. CSCs reside in a niche where immune cells, microvesicles, and cytokines can stimulate self-renewal and promote metastasis. These microvesicles and cytokines may also exert immunosuppressive functions. Some studies have found that CSCs can weaken T-cell differentiation, proliferation, and anti-tumor effects [48]. The CD44 molecular structure performs as a receptor for hyaluronic acid and other components of the ECM, enabling CSCs to identify changes in the environment and adapt their effects according to those changes and the new environment [45]. Indeed, CD44 is a critical cell surface molecule capable of detecting, integrating, and transducing signals from the cell microenvironment to cytoskeletal proteins or the nucleus to regulate a range of gene expressions that govern cell behavior. Aberrant expression and dysregulation of CD44 contribute to tumor initiation, progression, and metastasis development. Therefore, CD44 overexpression represents an unfavorable prognostic factor for CRC patients and could be used to predict tumor-associated poor prognostic features.

Several studies have revealed the central role of high CD44 expression in carcinogenesis, tumor growth, differentiation, and tumor metastasis in CRC [49,50,51,52]. In a study by Weber et al., knockout of the CD44 gene prevented tumor metastasis, even though the tumor continued to exist in mice [48]. Du et al. found that a single CD44-positive cell from a CRC tissue can grow to form a mass of tissue similar to a primary tumor [23,53]. The results of our study are substantially in line with literature data. Our analysis showed that patients with overexpressed CD44 have a worse prognosis in terms of overall survival.

Furthermore, our work demonstrated the association of high CD44 expression with clinically poor prognostic features. Some previous research reported that the expression level of CD44 was higher in high-grade CRCs than in low-grade tumors, and this overexpression was associated with reduced patient survival [53,54]. In line with that study, we also found that patients overexpressing CD44 more frequently had a high-grade tumor than patients with low CD44 expression. In this regard, there are conflicting data in the literature; some studies showed that loss of CD44 expression was associated with increased tumour aggressiveness [55,56]. This inconsistency in results may be due to alternative splicing of CD44 pre-RNA. Higher expression of CD44 in advanced tumors has been recorded in several types of malignancies, such as in lymphomas, and gastric and cervical carcinomas, possibly associated with poor prognosis [57,58]. Consistent with these data, increased expression of CD44 also was found to correlate with advanced tumor stage in the current study.

Moreover, in previous studies, a statistically significant association was found between positive CD44 expression and left-sided tumors in an Egyptian population [59], which tended to be associated with a better prognosis than right-sided CRC localization [60,61]. In the current study, there was no significant association between CD44 and tumor site, which could be due to the difference in the number and type of patients.

Similar to the literature data, our analysis confirms the association between CD44 expression and poor prognosis. However, our study documented that differential CD44 expression impacts patient prognosis differently. This finding is certainly new. We thus obtained that mild or moderate expression of CD44 is not associated with a worse prognosis in the same way as no expression. On the contrary, high expression is associated with a significantly worse prognosis. Furthermore, high CD44 expression is also associated with poor clinical prognostic features.

Interestingly, we used a simple and inexpensive method, immunohistochemistry. Specifically, we used an already known and established expression scoring scale for HER2 in breast cancer. We assessed the number of cells expressing CD44 and the intensity of CD44 expression in the different samples during immunohistochemical evaluations. We realized that the variability within our case series was much more related to the intensity of staining than the percentage of positive cells. This observation led us to choose a scoring system based on the HER2/neu scheme. The choice of this score was positively correlated with the patient’s prognosis and clinical–pathological features. For this reason, it might play a role in identifying poor prognosis mCRC patients.

Finally, many studies have evaluated the different isoforms in the literature or used very complex methods that are difficult to implement in standard clinical practice. For these reasons, the results of our work and the methodology used appear innovative.

Concerning first- and second-line responses in terms of PFS and RR, our study showed no statistically significant difference in patients with high and low CD44 expression. However, we can see a trend towards higher treatment efficacy in patients with low CD44 expression compared to those with high expression. The lack of statistical significance could be related to the study’s retrospective nature and a poorly selected and non-homogeneous population. The patients involved in our analysis received first- and second-line chemotherapy treatment according to each patient’s clinical characteristics, the tumor’s biological features, and the physician’s preferences. There is substantial uniformity between the treatment used in low and high CD44 expression mCRC patients. However, it is useful to note a trend towards greater first-line triplet use in highly expressed CD44. This is likely related to the poor clinical features in this subgroup. Furthermore, the regimens used did not impact the differences in outcomes between patients with high and low CD44 expression.

Therefore, it is difficult to draw definitive conclusions on resistance to anti-cancer treatments. However, the trend towards a better response in patients with low CD44 expression, together with data on overall survival and solid biological basis, suggests that CD44 could be used not only as a marker of poor prognosis but also of chemoresistance. Prospective studies will help confirm the role of CD44, as a marker of both poor prognosis and poorer response to anti-cancer treatments. Moreover, due to the ease and low cost of immunohistochemical analysis, CD44 could be an excellent marker of poor prognosis and chemoresistance in our clinical practice.

Due to its retrospective nature, this study carries some critical issues as it included the analysis of the outcomes of a relatively small number of patients, and its results should be handled with caution. To obtain more robust results, it will be useful to prospectively evaluate the expression of CD44 in a larger sample size of CRC patients. In this context, the analysis of a more uniform population could be helpful, making more information available on the importance of the molecular profile and treatments performed on the patient’s prognosis.

Finally, in light of the above, we think that CD44 research may have further fields of use. In fact, several pre-clinical and clinical studies have been conducted to evaluate the pharmacokinetics, efficacy, and toxicity of anti-CD44 therapies in CD44-expressing tumors. The main categories of CD44-targeted therapies currently under investigation include neutralizing antibodies, mimetic peptides, aptamers, natural compounds that suppress CD44 expression, direct targeting of HA by bioconjugates and nanoparticles, HA oligomers, and CD44 decoys [62]. The data from these studies seem encouraging from both an efficacy and safety perspective. Indeed, these studies support the continued development of therapeutic strategies to target CD44 expression in patients. However, it remains paramount to investigate the possible short- and long-term consequences of CD44-directed drugs in future studies, given the numerous implications of CD44. These include its role in T-cell activation [63].

## 5. Conclusions

In conclusion, our work confirmed the prognostic value of CD44 expression in CRC. Specifically, high CD44 expression was significantly associated with higher proliferative activity of CRC and poor prognosis. Furthermore, CD44 overexpression was also associated with clinically poor prognostic features: older age, inoperable disease, stage IV at diagnosis, mutated BRAF, and high-grade tumor. Therefore, our results suggest that CD44 induced by malignant cells might downregulate tumor cells’ apoptosis, leading to a decline in growth inhibition and increased ability to resist chemotherapy. However, in our analysis, it is difficult to draw definitive conclusions on resistance to cancer treatments. Given the tendency for a better response in patients with low CD44 expression, the data on overall survival, and the solid biological basis, our work suggests that CD44 could also be a predictive marker of chemoresistance. Future prospective studies on larger sample sizes are needed to confirm our results and investigate the possible predictive role of resistance to anti-cancer therapy. Finally, in addition to its role as a predictive biomarker of poor prognosis and an independent predictor of tumor development, CD44 may be an effective therapeutic target. Potential therapeutic strategies targeting CD44-positive tumors, effectively blocking CD44, might provide ample opportunities to overcome chemoresistance in various cancer types.

## Figures and Tables

**Figure 1 cancers-15-01212-f001:**
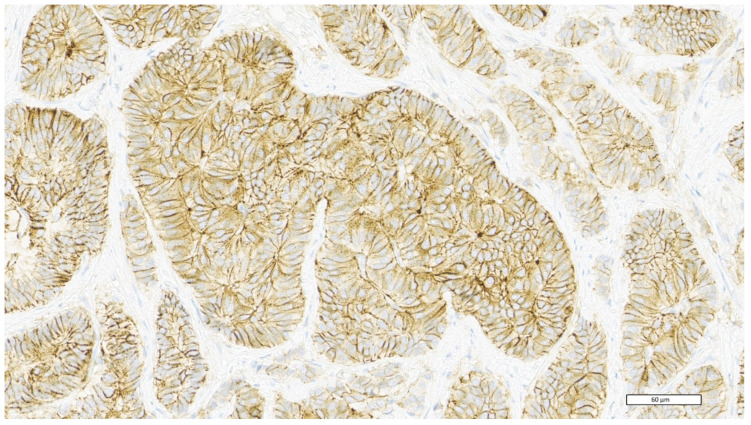
CD44 high expression—score 3+: intense membrane staining in at least 10% of tumor cells.

**Figure 2 cancers-15-01212-f002:**
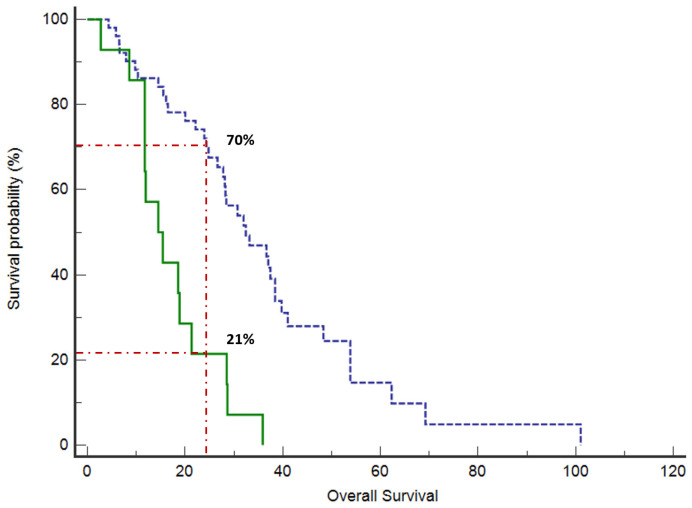
Median OS was 30.7 months (95%CI 27.8 to 101) in patients with low CD44 expression (blue line, dashed) versus 14.5 months (95%CI 11.7 to 35.9) in high CD44 expression group (green line, continuous) (*p* = 0.0001). The proportion of patients alive at 24 months was 70% in the low CD44 expression group versus 21% of high CD44 expression patients (*p* = 0.0001).

**Table 1 cancers-15-01212-t001:** CD44 scoring system.

CD44 Expression	Score
Negative or weak membrane staining in less than 10% of tumour cells	0
Weak membrane staining in at least 10% of tumour cells or moderate membrane staining in less than 10% of tumour cells	1+
Moderate membrane staining in at least 10% of tumour cells or intense membrane staining in less than 10% of tumour cells	2+
Intense membrane staining in at least 10% of tumour cells	3+

**Table 2 cancers-15-01212-t002:** Patient’s characteristic. Abbreviations: ECOG PS = Eastern Cooperative Oncology Group performance status; K-RAS = Kirsten rat sarcoma viral oncogene homologue; N-RAS = Neuroblastoma RAS viral oncogene homologue B1; Anti-VEGF: anti-vascular endothelial growth factor antibodies; Anti-EGFR: anti-epidermal growth factor antibodies.

	Low CD44 Expression	High CD44 Expression
Gender M F	28 (54.9%) 23 (45.1%)	11 (78.6%) 3 (21.4%)
Age <70 ≥70	39 (76.5%) 12 (23.5%)	8 (57.2%) 6 (42.8%)
Stage at diagnosis I–III IV	20 (39.2%) 31 (60.8%)	1 (7.1%) 13 (92.9%)
CDX-2 Positive Negative	48 (94.1%) 3 (5.9%)	11 (78.6%) 3 (21.4%)
Site of primary tumour Sx Dx	38 (74.5%) 13 (25.5%)	10 (71.4%) 4 (28.6%)
Surgery of the primary tumour Yes No	46 (90.2%) 5 (9.8%)	7 (50%) 7 (50%)
Tumour Grade Good–moderate Poor	41 (80.4%) 10 (19.6%)	6 (42.8%) 8 (57.2%)
Metastases sites Single site Multiple site	19 (37.3%) 32 (62.7%)	3 (21.4%) 11 (78.6%)
Liver Metastases Yes No	34 (66.7%) 17 (33.3%)	12 (85.7%) 2 (14.3%)
Peritoneal Metastases Yes No	16 (31.4%) 35 (68.6%)	6 (42.9%) 8 (57.1%)
K-RAS/N-RAS mutational status Wild type Mutant	27 (52.9%) 24 (47.1%)	7 (50%) 7 (50%)
B-RAF mutational status Wild type Mutant	46 (90.2%) 5 (9.8%)	9 (64.3%) 5 (35.7%)
First-line chemotherapy Monotherapy Doublet Triplet	6 (11.8%) 42 (82.3%) 3 (5.9%)	2 (14.3%) 8 (57.1%) 4 (28.6%)
First-line biological drug Anti-VEGF Anti-EGFR None	32 (62.8%) 7 (13.7%) 12 (23.5%)	9 (64.3%) 2 (14.3%) 3 (21.4%)
Second-line chemotherapy Monotherapy Doublet None	3 (5.9%) 32 (62.7%) 16 (31.4%)	1 (7.1%) 7 (50%) 6 (42.9%)
Second-line biological drug Anti-VEGF Anti-EGFR None	22 (43.1%) - 29 (56.9%)	5 (35.7%) 1 (7.1%) 8 (57.2%)

**Table 3 cancers-15-01212-t003:** Response to first and second line. Abbreviations: ORR = overall response rate; DCR = disease control rate; CR = complete response; PR = partial response; SD = stable disease; PD = progression disease; PFS-1 = progression-free survival in first line; PFS-2 = progression-free survival in second line; mo = months.

Response	Low CD44 Expression	High CD44 Expression	*p*-Value
ORR first line, *n* (%) ORR second line, *n* (%)	21 (41.2%) 4 (12.1%)	6 (42.9%) -	*p* = 0.9 *p* = 0.3
DCR first line, *n* (%) DCR second line, *n* (%)	43 (87.8%) 26 (78.8)	10 (71.4%) 4 (50%)	*p* = 0.14 *p* = 0.1
Median PFS-1, mo (range)	11.6 mo (7 to 93.5)	9.5 mo (3.9 to 19.4)	*p* = 0.17
Median PFS-2, mo (range)	5.6 mo (3.5 to 26)	2.9 (2.1 to 9.6)	*p* = 0.12

**Table 4 cancers-15-01212-t004:** Correlation between high CD44 expression and clinically poor prognostic features.

Variable	OS—*p*-Value
Age (≥70 y)	0.0166
CDX2 negative	0.09
Stage IV at diagnosis	0.0241
Right-sided colon cancer	0.8
Inoperable colon cancer	0.0008
High-grade tumour	0.0084
Metastasis sites (single vs. multiple sites)	0.27
Liver Metastases	0.16
Peritoneal Metastases	0.4
B-RAF mutational status	0.0111

## Data Availability

Datasets generated during and/or analyzed during the current study are available from the corresponding author on reasonable request. The data are not publicly available due to privacy restrictions.

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
