# Peer review of "New Horizons in Metastatic Colorectal Cancer: Prognostic Role of CD44 Expression"

_cancers, 2023, doi:10.3390/cancers15041212_

Round 1

Reviewer 1 Report

In this study, Ziranu et al. explore the association between CD44 expression and the clinicopathological 17 features and survival of metastatic colorectal cancer (mCRC) patients. They observed that overexpressed CD44 was associated with poor prognosis, with a median survival of 14.5 months versus 30.7 months in low CD44 expression. Furthermore, CD44 high expression was also associated with clinically poor prognostic features. These are certainly interesting findings and align with many CD44-related existing studies in different tumors. However, the authors need to address the following concerns to make the article acceptable.

  1. The major concerns of this study are novelty and sample size. The prognostic potential of CD44 for CRC has already been established in several studies (e.g., PMID: 31114754, PMID: 27323782, etc.). The authors need to provide additional evidence for addressing this concern of novelty. The authors also need to improve the sample size substantially.
  2. In colon cancer, is CD44 only expressed in stem cells? Does the abundance of staining in stage IV indicate an increase in stem cells?
  3. The authors conclude that CD44 targeting could be useful for overcoming chemoresistance. However, CD44 is a critical marker of T cell development as well as effector memory T cells. It would further expose the patients to other diseases.
  4. Based on their findings, the conclusion needs regarding CD44 expression needs to be clarified. Does CD44 increase with the age of the disease, or does CD44 expression drive the stage of cancer? 
  5. What was the treatment regimen of these patients? Can that affect CD44 expression?

Minor:

  1. Figure 1 labeling in the text does not match the statement. It needs a separate sentence describing the figure and explaining the result in the following statement.
  2. A recurring concern throughout the article is abbreviations. There are so many words (e.g., patients, membrane staining, etc.) that are unnecessarily abbreviated. These are not common abbreviations. Furthermore, many abbreviations (e.g., TTP) haven't been defined. The authors need to delete the unnecessary abbreviations and define all that are kept.

Reviewer 2 Report

This article describes the different levels of CD44 protein in 65 patients suffering from metastatic colorectal cancer. Although the population of study is sufficient and the study concept is valid, the methodology is oversimplified and is not enough to support the conclusions drawn by the authors - yet it can be easily fixed. 

Here are the detailed suggestions for current article:

Title: this article does not really describe the role of CD44 expression, as the role is already (at least basicly) explained and further investigation of the CD44 role would require much more effort and different study design. I suggest changing the title to actually correspond with the article's finding, including potential as a prognostic factor or different expression in high grade CRC. 

Methods: it's the most important issue. Comparison of expression of CD44 was based solely on arbitrary expression strength evaluation and almost (besides score 0) does not take into consideration the number of cells expressing protein of interest. It is oversimplified and makes the conclusions drawn unreliable. I suggest to use one of those solutions:

- IRS scale (developed here: Remmele, W.; Stegner, H.E. Recommendation for uniform definition of an immunoreactive score (IRS) for immunohistochemical estrogen receptor detection (ER-ICA) in breast cancer tissue. Pathologe 19878, 138–140 and used and explained for example here: https://iv.iiarjournals.org/content/invivo/34/1/213.full.pdf). It is easy to use, semi-quantitative scale, which will give more reliable results, as it takes into consideration not only the intensity, but also extent of reaction. 

- Image analysis software, such as opensource QuPath or different, which allows to automatically evaluate the intensity of reaction and number of positive cells, is free and easy to use. Also - less prone to observer bias. 

It's up to authors which one methods they will apply. 

Results: Lines 179-181: Generally, if there is statistically insignificant difference, you can't say about any difference at all - as they were far from being statistically significant, there is quite a chance that the differences just happened by accident and are not a real thing - that's why statistical inference is used. Therefore, you can't say that there is a difference, but it's not statistically insignificant - but that there is no statistically significant difference. 

Introduction or Discussion sections do not raise significant concerns - yet the discussion and conclusions drawn must be changed, if IRS/image analysis software will give different results than previously used oversimplified methods.

Round 2

Reviewer 1 Report

The authors have addressed the majority of my comments. The only minor concern I still have is the sample size of their study. The authors can add a small section on the caveats of their current study and one of the points they can mention is the sample size.
